# Common Challenges and Identified Solutions for State Newborn Screening Programs during COVID-19 Pandemic

**DOI:** 10.3390/ijns8010007

**Published:** 2022-01-18

**Authors:** Dylan Simon, Elizabeth Broadbridge, Mei Baker, Amy Gaviglio, Dorota Gruber, Kimberly Noble Piper, Norma P. Tavakoli, Jamie Sullivan, Annie Kennedy

**Affiliations:** 1EveryLife Foundation for Rare Diseases, Washington, DC 20005, USA; lbroadbridge@everylifefoundation.org (E.B.); jsullivan@everylifefoundation.org (J.S.); akennedy@everylifefoundation.org (A.K.); 2Newborn Screening Laboratory, Wisconsin State Laboratory of Hygiene, University of Wisconsin School of Medicine and Public Health, Madison, WI 53706, USA; mei.baker@slh.wisc.edu; 3Genetics and Metabolism Division, Department of Pediatrics, University of Wisconsin School of Medicine and Public Health, Madison, WI 53792, USA; 4Center for Human Genomics and Precision Medicine, University of Wisconsin School of Medicine and Public Health, Madison, WI 53705, USA; 5G2S Corporation/CDC Newborn Screening and Molecular Biology Branch, Atlanta, GA 30341, USA; agaviglio@cdc.gov; 6Department of Pediatrics, Cohen Children’s Medical Center, Northwell Health, New Hyde Park, NY 11040, USA; dgruber1@northwell.edu; 7Departments of Pediatrics and Cardiology, Donald and Barbara Zucker School of Medicine at Hofstra/Northwell, Hempstead, NY 11549, USA; 8Center for Congenital and Inherited Disorders, Iowa Department of Health, Des Moines, IA 50319, USA; Kimberly.piper@idph.iowa.gov; 9Division of Genetics, Wadsworth Center, New York State Department of Health, Albany, NY 12201, USA; norma.tavakoli@health.ny.gov; 10Department of Biomedical Sciences, State University of New York, Albany, NY 12222, USA

**Keywords:** newborn screening, COVID-19, pilot studies, follow-up services, public health laboratories

## Abstract

During the COVID-19 pandemic, state newborn screening programs faced challenges to ensure this essential public health program continued to function at a high level. In December 2020, the EveryLife Foundation for Rare Diseases held a workshop to discuss these common challenges and solutions. Newborn screening officials described challenges including short staffing across the entire program, collection and transport of specimens, interrupted follow-up activities, and pilot study recruitment. To address these challenges, state programs implemented a wide variety of solutions to maintain the high standards of newborn screening. To address staffing issues, newborn screening programs, public health laboratories, and hospitals all cross-trained personnel, worked to manage staff stress, and established essential functions. Other solutions included working with courier companies to ensure the timely pick-up of specimen, creating educational materials for hospital staff, and the creation of hybrid recruitment models for pilot studies. Implementing the lessons discussed throughout this paper can help to prepare for the next public health emergencies to ensure that a program that interacts with millions of families every year and saves the lives of thousands of children every year is minimally impacted.

## 1. Introduction

In December 2020, the EveryLife Foundation for Rare Diseases held a workshop to identify challenges that have arisen in state newborn screening (NBS) programs nationwide during the COVID-19 pandemic, as well as to discuss solutions implemented by multiple rare disease stakeholders. The rare disease community includes 25–30 million Americans, about 50% of whom are children [1]. These Americans have at least one of more than 7000 rare diseases with varied causes, symptoms, progression, and prognosis [2]. Many individuals with rare diseases are considered significantly immunocompromised and others have severe heart, lung, kidney, or liver damage due to their rare disease. Patients and their families have largely put their lives on hold, making all possible sacrifices to reduce the likelihood that they would not contract COVID-19 due to their pre-existing health challenges. 

The COVID-19 pandemic strained the U.S. healthcare system, limiting the resources available to those who previously relied heavily on the system. This impact stretched across every public health agency in the country, including state NBS programs. NBS, which is considered one of the most successful public health programs in the country, tests the approximately 3.84 million newborns born in the United States for a variety of rare diseases that can cause serious health problems starting in infancy or childhood [3]. Early detection and treatment can help to prevent intellectual and physical disabilities for the life-threatening diseases found on NBS panels [4]. State NBS programs use a multi-faceted approach to ensure that specimens from each of the almost four million babies born every year are collected, tested, and results properly reported. State program’s responsibilities include more than just the test itself, with programs overseeing the education about conditions for parents, training for health care professionals, follow-up services to connect families that require confirmatory testing with specialty providers, and quality assurance testing for the NBS test process [5]. Each part of the system plays a vital role in ensuring the success of NBS. 

Issues that arose during the COVID-19 pandemic may occur during any public health emergency. Identifying gaps in operations and possible solutions utilized during the pandemic will help to prepare for the next public health emergency. NBS is an essential program, and each state must work to ensure its ability to function at a high level during a public health emergency. During the pandemic, each state’s NBS program experienced similar issues at each stage of the NBS process, including collection of specimens, testing of specimens, and follow-up services (Figure 1). This meeting report will focus primarily on state-based issues from representatives of three states (NY, IA, WI) presenting at the EveryLife Foundation for Rare Diseases 2020 Scientific Workshop. The issues and challenges of conducting NBS during the pandemic have been reported by other state NBS programs as well and the three state presentations serve as case examples. Of note, identified issues and challenges are not limited to the ones discussed in the following sections, and other state NBS systems may well have found alternative/additional solutions [6]. 

## 2. Materials and Methods

The challenges and solutions discussed in this paper were collected from presentations at the EveryLife Foundation for Rare Diseases 2020 Annual Scientific Workshop on 15 December 2020. Speakers were chosen through both an open call to for abstracts, as well as, a proactive identification of programs that could provide relevant experiences. The single day, virtual event was split divided into multiple sections, with each consisting of multiple presentations and a question and answer session. This paper focuses on the newborn screening section of this workshop.

## 3. Results

### 3.1. Staffing Issues

NBS programs across the country faced difficult staffing decisions to accommodate the challenges that arose during the pandemic. Staff furloughs, sick or quarantined staff, or staff reassignment impacted hospitals, state public health laboratories (PHL), and follow-up staff [6]. Short staffing resulted in extra work in day-to-day tasks, including COVID-19 testing, isolation of COVID-19 positive staff, parent support, and reporting requirements [6]. In hospitals and NBS programs in the three states, staff were required to undergo COVID-19 specific training and to complete daily COVID-19 symptom monitoring, including temperature checks and self-administered COVID-19 symptoms-monitoring questionnaires. Team members who experienced symptoms of COVID-19 or were exposed to a COVID-19 positive individual were required to undergo testing to rule out the possibility of COVID-19 infection. CDC recommended personal protective equipment (masks, goggles, shields, and gowns when necessary) be worn while on the hospital premises. Social distancing measures were implemented throughout the hospital and public health systems. In each state Public Health Laboratories (PHLs), decisions were made regarding which work functions were essential, which could be performed remotely, and what requirements were necessary to maintain efficacy. Most of the work performed by laboratory staff, including opening mail and accessioning specimens and testing, had to be performed in the laboratory as opposed to their normal location. Staff had to wear masks and follow the social distancing rules devised by the organization. Staggered work schedules were established and staff performing functions such as data entry and follow-up worked from home, when possible, to reduce density at work and reduce the possibility of staff infections. 

To minimize patients’ in-hospital exposure, early discharges were implemented in New York, which required prioritization of tasks and major/frequent adjustments to the staffs’ day-to-day workflow. In addition, early discharges often meant that the NBS specimen was collected earlier than recommended, necessitating repeat specimen collection and more clinical follow-up. In some cases, either staff were given the opportunity, or were required, to work over-time and assist in the pandemic response, in addition to their routine work. The frequent changes to the routine and additional work over a prolonged period have led to physical and emotional exhaustion and mental health issues. Studies have shown that during the pandemic healthcare workers reported work overload and increased stress, anxiety, and burnout [7]. The stress caused by the pandemic and the additional work causing burnout in staff were factors that needed to be addressed in a timely fashion. Prevention strategies were put in place to assist staff to cope in healthy ways with the stresses caused by the pandemic. Frequent and constant communication between staff and staff and leadership was encouraged to help cope with stresses and unpredictability caused by pandemic related factors. 

Hospitals and PHLs in each state implemented a wide variety of solutions with a few common solutions found across multiple programs. Solutions such as temperature checks and mask-wearing were included in all COVID-19 protocols. NBS programs implemented staff cross-training to prepare for additional and new work, including training nurses on how to collect necessary specimens and training non-NBS PHL staff on the functions and operations of the NBS lab. In addition, remote work was established where possible, determining the minimum number of staff needed for daily essential tasks that required in-person work and accommodating those tasks. 

For follow-up services, staffing issues impacted the ability to arrange needed visits for repeat specimen collection or diagnostic evaluation. After a positive result on NBS, hospital or NBS program staff normally facilitate follow-up visits to perform confirmation testing with primary care providers (PCPs) or specialty care providers for the at-risk child. In Iowa, all short-term follow-up staff contributed to the COVID-19 response at the hospitals, limiting the number of staff available to arrange these follow-up visits. In addition, staff shortages at primary care offices resulted in NBS staff waiting between 10–40 min on the telephone before being able to speak to someone. On select occasions, NBS staff was unable to contact PCPs altogether, resulting in crucial follow-up visits not occurring. The Iowa state PHL decided to make arrangements directly with families for repeat and confirmatory testing rather than relying on the PCPs to schedule or collect the specimen due to the volume of work PCPs experienced which made it difficult for them to make the proper arrangements.

In New York, staff employed by the State were allowed to increase the amount of banked vacation leave. This encouraged staff who would otherwise have used up a portion of their vacation leave prior to the yearly deadline to postpone vacation time to a later date. This State policy permitted more flexibility for staff and prevented understaffing during periods when staff would otherwise have taken time off.

### 3.2. Collection of Specimens

The NBS process begins with the time-sensitive collection and shipment of specimens to the PHL for testing. In the U.S., the recommendation is that specimens be collected between 24 and 48 h of age [8]. Many NBS programs consider specimens collected at < 24 h of age as not optimal for testing for disorders such as congenital hypothyroidism, congenital adrenal hyperplasia, and cystic fibrosis, due to elevated levels of thyroid stimulating hormone, 17-hydroxyprogesterone and immunoreactive trypsinogen, respectively, during the first 24 h after birth. During the pandemic, there was a concern by NBS programs that NBS specimens may be collected prior to 24 h to accommodate the earlier discharge of mothers and babies from hospitals. Therefore, cut-offs for some analytes were re-evaluated. In addition, the verbiage on the New York NBS program reports was modified to collect repeats for suboptimal specimens and borderline specimens as soon as practical.

The courier system is a vital part of NBS timeliness, ensuring that the specimens are delivered to the laboratory in a suitable time frame. During the pandemic, courier staff in all three states experienced multiple issues such as not being allowed at standard pick-up locations within the hospital, general transport delays due to increased shipping demands nationwide, and increased COVID-19-related pickups. In Iowa, the inclusion of the director of the courier service as a partner in solving transport issues went a long way to determine the resources needed to continue operations. In addition, the courier services worked directly with hospitals to identify other locations/options for specimen collection to ensure they were still picked up and delivered to the PHL. 

### 3.3. Testing at the Public Health Laboratory

State PHLs faced multiple challenges to ensure that the COVID-19 pandemic did not interrupt their work. Common challenges included the staffing issues discussed above as well as supply shortages and a shift to remote work.

PHLs in Iowa and New York implemented remote work to limit the amount of time the staff spent in the lab. The remote work allowed staff to conduct data entry, data analyses for reporting, and follow-ups from home, to help comply with pandemic protocols. Information technology (IT) availability within PHLs was severely limited due to IT staff being pulled away to COVID-19-related testing and data needs, which resulted in delays in implementation of needed IT elements like new laboratory information management systems (LIMS) or LIMS enhancements to add new diseases within NBS programs. 

In New York, laboratory staff were cross-trained and a trial for 6-day (Saturday) testing was launched. Furthermore, remote connection to laboratory instruments was established so that some functions (e.g., data review) could be performed remotely. In addition, some staff volunteered to assist with accessioning COVID-19 specimens prior to receipt of NBS specimens on weekdays and on Saturdays, leading to a longer work week. 

In Wisconsin, the PHL saw similar challenges and implemented multiple solutions to maintain their work. NBS was deemed an essential service, which ensured that the PHL was not reassigned to COVID-19 testing and ensured that the lab was supported throughout the pandemic. Their work to adapt to the pandemic prevented any possible delays in results reported within first week of life, suggesting that the mitigation measures prevented the pandemic from impacting the ability of the newborn screening laboratory.

NBS for approximately 3.84 million newborns in the U.S. each year for a multitude of disorders requires a vast number of supplies, reagents, kits, and consumables as well as various chemicals. During the pandemic, issues with production and transportation led to delays in receiving needed supplies. An added complication was that NBS laboratories were competing with laboratories performing COVID-19 testing for supplies such as gloves, tips, and PCR plates. To ensure uninterrupted testing during the pandemic, NBS programs had to ensure effective communication with their suppliers and find alternative sources, if needed. In some instances, NBS programs shared supplies amongst laboratories to assist programs that had limited availability.

### 3.4. Follow-Up Activities

In New York, some of the specialists who, under normal circumstances, evaluated and treated the newborns with abnormal results, were reassigned to assist with their institutions’ COVID-19 response, limiting their ability to perform their regular NBS duties. In response, the NBS program created educational packages for PCPs to assist them in educating families regarding follow-up of abnormal results and to ensure appropriate confirmatory testing was ordered. In addition, all specialty care centers in New York were alerted to the changes made to the NBS process, and updated emergency contact information was requested from them so that they could easily be reached in case of an emergency result. Telehealth services rolled out by the hospitals were utilized to help communication between parents and providers. 

Under ordinary circumstances, babies with borderline results, or whose specimens are of poor quality, need an additional NBS specimen collected and submitted for screening. In New York, a repeat screen is often collected by the birth hospital, and less frequently, by the pediatrician. A mass email was sent to the PCPs in New York to request their assistance in collecting repeats, as many hospitals closed their doors to any outpatient visits, some were converted to COVID-19-only facilities, and other hospitals faced staffing shortages and reassignments. In addition, parents feared exposing their newborns to the risks associated with entering health care facilities during a pandemic unless for an emergency. During the spring of 2020, at the height of the first wave of the pandemic, a larger number of cases were closed as “lost to follow-up” because no repeat screen was received by the New York NBS Program secondary to the above concerns.

### 3.5. Newborn Screening Research 

In addition to administration of routine NBS, some PHLs also participate in NBS research and program improvement activities to support efforts to add new conditions to NBS panels. An important step in adding a new condition to the list of diseases routinely screened is a pilot study to assess feasibility and utility of screening for that condition. This work faced significant setbacks throughout the COVID-19 pandemic and the solutions to many of these challenges discussed below can be used to inform best practices for future public health emergencies. 

Led by Parent Project Muscular Dystrophy, a consortium of organizations including the New York State NBS Program and ACMG NBSTRN, in collaboration with Northwell Health and New York Presbyterian Health Systems, performed a consented pilot study to screen newborns for Duchenne muscular dystrophy, recruiting newborns from October 2019 to September 2021. The pilot was designed to validate a high-throughput immunoassay screen for Duchenne and create a follow-up pipeline for molecular diagnostic testing. The goal of this pilot was to identify infants with Duchenne in the newborn period, prior to symptom-onset, and allow confirmation of diagnosis, identification of the Duchenne genotype and determine treatment course, including potential participation in clinical trials. 

The protocol for the Duchenne pilot study and any changes to the protocol due to the COVID-19 pandemic were approved by the institutional review boards of the institutions involved in the study. Study staff at the laboratory and at hospitals very quickly determined ways to adapt to the necessary limitations imposed by the pandemic and determined practical workarounds [9]. 

The recruitment for the pilot study began in October of 2019 and accelerated between January and February of 2020 as recruitment staff were hired and trained. The original study design included an in-person informed consent process that involved bedside recruitment of eligible participants. However, hospitals suspended in-person research activities and the study team had to decide on suspending the study or adapting to the COVID-19 pandemic by implementing remote recruitment. Remote recruitment (by phone and online) started in mid-March 2020 and allowed for enrollment to continue despite reduced or, in some cases absent, onsite personnel. However, the method posed several logistical challenges, including increased workload and follow-up for staff, issues with computer illiteracy requiring extensive instructions (for a subset of families), late consent forms, and, importantly, delays in reaching eligible patients until after routine NBS had been completed. Although remote recruitment did not result in recruitment of the levels reached by in-person recruitment, it did allow the study to continue, and it allowed for the infrastructure to be put in place to perform hybrid (both in-person and remote) recruitment. 

The hybrid approach began in July of 2020. During the hybrid approach, participant recruitment was completed on-site, and some families who were discharged without being recruited were contacted by telephone or email within the first 4 weeks of life, and then recruited often after the routine NBS panel was completed. In addition, patients were also given the option of using paper mailed-in consent forms. This approach resulted in a significant increase in enrollment with numbers exceeding those in peak pandemic and pre-pandemic months [9]. 

For the pilot study, testing for Duchenne was a two-step process with the initial screening at the NBS program requiring a DBS specimen and the second-tier molecular test requiring whole blood. During the pandemic, telehealth follow-up visits were offered to families whose newborns had been referred for follow-up to help with compliance with genetic counseling and confirmatory molecular testing visits. Remote sample collection strategies were also implemented. To minimize visits to health care settings, parents were given the option to collect a specimen for second-tier testing at home. A sample collection kit was mailed to them, and they were able to collect a buccal swab and submit it directly to the molecular testing laboratory. In addition to finding success with the hybrid approach to study recruitment, the research team noted several other ways they tackled the challenges of the COVID-19 pandemic. Like the Wisconsin PHL, the research team kept in contact with regional suppliers to prepare in advance for shortages as much as possible. Similar to other groups, the team took advantage of remote data entry and underscored the need for a flexible Research and Electronic Data Capture (REDCap), LIM system and IT support [10].

## 4. Conclusions

The COVID-19 pandemic is one of the greatest public health emergencies this country has faced. However, it is important to recognize that public health emergencies will continue to occur beyond this pandemic, from smaller disease outbreaks to public health emergencies associated with natural disasters. NBS in Iowa, New York, and Wisconsin faced common and unique challenges during the pandemic, working to find solutions to ensure that this critical program continues to run properly (Table 1). Lessons learned during the COVID-19 pandemic can help to prepare for the next public health emergencies to ensure that NBS is minimally impacted. 

A key lesson learned was the importance of designating NBS as an essential service and its workers as essential workers. The Wisconsin NBS program received strong support from the state government and the Wisconsin State Laboratory of Hygiene leadership from the outset of the pandemic, which played an important role in ensuring that the NBS laboratory outcomes were not impacted. NBS helps thousands of families a year receive a diagnosis that allows affected newborns to begin health-saving treatment when it makes the biggest impact [4]. During a public health emergency, designating newborn screening as an essential service will ensure that those diagnoses continue with minimal obstacles.

In-person hospital visits are difficult in any public health emergency. While the COVID-19 pandemic shortened hospital visits and restricted hospital visitations, other public health emergencies, such as hurricanes or wildfires, may directly interrupt the ability for patients to get to a hospital. It is important for NBS programs to identify ways to conduct follow-up services with families outside of the standard follow-up through PCPs. During the COVID-19 pandemic, Iowa worked directly with patients to identify how to set up confirmatory testing, while New York implemented a hybrid model (Telehealth and in-person visit) to ensure patients would comply with the follow-up testing. A hybrid model was also implemented by New York hospitals participating in the Duchenne pilot, to ensure successful recruitment to the Duchenne pilot. These changes are just a few options state NBS programs can consider in preparation for of the next public health emergency. The growth of telehealth will continue to help in this area. While diagnostic testing will still require in-person visits, telehealth will help PHLs connect with families that require confirmation testing and facilitate in-person testing. Additionally, in some cases, in-home specimen collection kits can be used by parents to submit specimens for confirmatory testing (e.g., buccal swab collection kits for molecular testing).

The stress of the pandemic on healthcare workers has been documented and must be researched further [4]. In addition, interventions that can assist workers to mitigate the effects of pandemic-related stress and increased workload should be investigated. For organizations to function effectively through future pandemics emphasis must be placed on the morale and mental health of staff.

The COVID-19 pandemic dramatically changed the ways our public health systems operate. The workshop provided an opportunity to identify lessons learned during that time, particularly throughout the NBS ecosystem. Implementing the lessons discussed throughout this paper will help to continue to improve a system that interacts with millions of families every year and saves the lives of thousands of children every year.

## Figures and Tables

**Figure 1 IJNS-08-00007-f001:**
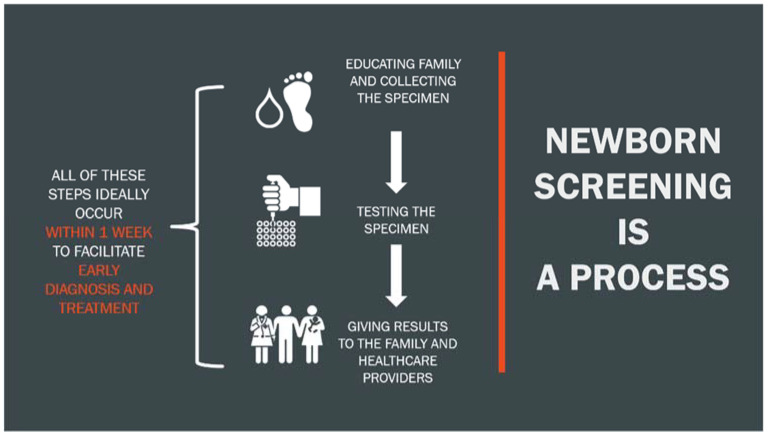
Newborn screening process following a birth.

**Table 1 IJNS-08-00007-t001:** Common Challenges faced by state NBS programs during the COVID-19 Pandemic and Identified Solutions.

Challenge	Identified Solution
Short Staffing (staff furloughs, sick or quarantined staff, staff reassignment)	Identifying essential functions, staggered work schedules, and staff cross-training
Staff Stress	Assist staff to cope in healthy way, frequent virtual team meetings to keep contact with each other during isolation and to address any issues that the team members were facing
Specimen Collection Timing with Early Hospital Discharges	Re-evaluate cut-offs and modifying protocol to collect repeats “as soon as possible”
Courier Pick-up	Work with couriers to determine a new pick-up location
Non-typical health care staff interacting with families regarding newborn screening	Create educational materials
Pilot Study Recruitment	Hybrid recruitment to include both remote and in-person methods
Collection of specimen for follow-up NBS research	Remote sample collection in combination with telehealth visits

## Data Availability

Recordings of the presentations are located at https://www.youtube.com/playlist?list=PLDScWcpkQ5CymgOPSEbKy0QUyohAvhf7y.

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
