# Peer review of "Common Challenges and Identified Solutions for State Newborn Screening Programs during COVID-19 Pandemic"

_2409-515X, 2022, doi:10.3390/ijns8010007_

Round 1

Reviewer 1 Report

The article summarizes a workgroup meeting to discuss challenges experienced by newborn screening programs during the early stages of the COVID-19 pandemic. Five key areas are discussed of which four are common to most newborn screening programs and one, research, is only performed by a subset of programs. The authors expand on specific challenges in these key areas and present solutions identified by the workgroup participants to address these challenges.

While the title and presentation of the article suggest that the identified challenges and solutions are broadly applicable across newborn screening programs, the findings are predominantly based on two large programs in the United States, Iowa and New York, with additional comments from Wisconsin. While the areas of staffing, specimen collection, testing, and follow-up were certainly common global challenges due to the pandemic, the way the identified programs responded are presented as common solutions to the issues rather than one way a very small subset of newborn screening programs, those who attended the workshop (NY, IA, WI), handled their response. In addition, a large portion of the results is dedicated to research, which is an important newborn screening activity but not one undertaken by most programs.  Developing the other four challenge areas; particularly collection, testing, and follow-up; similar to the section on research would be helpful. Finally, if continuity of operations plans (COOP) were discussed during the workshop, the paper would benefit from a discussion of that topic. If not, please disregard that suggestion.

Figure 1 has a typo in the top section with “N” missing from the word “SPECIMEN”.  Moreover, I find it curious that the authors included educating the family in the timeline of one week for the newborn screening process.  I believe it is widely agreed that education should start well before the collection of a specimen and suggest the authors consider a figure revision to reflect the importance of that step earlier in the process. On lines 129-130, the authors use the word “staff” twice in a sentence and it is unclear to which staff they are referring each time.

Author Response

Response to Reviewer 1 Comments

Point 1: While the title and presentation of the article suggest that the identified challenges and solutions are broadly applicable across newborn screening programs, the findings are predominantly based on two large programs in the United States, Iowa and New York, with additional comments from Wisconsin. While the areas of staffing, specimen collection, testing, and follow-up were certainly common global challenges due to the pandemic, the way the identified programs responded are presented as common solutions to the issues rather than one way a very small subset of newborn screening programs, those who attended the workshop (NY, IA, WI), handled their response.

Response 1: We appreciated this comment and made changes within the introduction in lines 77-83 of the manuscript to highlight the scope of the findings. The edits first highlight that the represenatives are only speaking about their three states. In addition, we made the edits to highlight that other state NBS systems may have found additional solutions and cite a location to find those solutions. In addition, edits were made throughout the manuscript to highlight that the solutions were ones drawn from the three states.

Point 2: In addition, a large portion of the results is dedicated to research, which is an important newborn screening activity but not one undertaken by most programs.  Developing the other four challenge areas; particularly collection, testing, and follow-up; similar to the section on research would be helpful.

Response 2: The newborn screening research discussed in the manuscript can lead to program improvements for state newborn screening programs who do not conduct research. While we agree with the reviwer that NBS research is not widespread, we believe that the discussion about remote sample collection, keeping in contact with regional suppliers, and how best to get in contact with families when they are not at the hospital can benefit newborn screening operations in states that are not conducting pilot study research. In terms of developing the additional challenge areas, the current content reflects the totality of the information discussed at the workshop. We included the reference highlighted above to provide an avenue for readers to read about additional solutions.

Point 3: Finally, if continuity of operations plans (COOP) were discussed during the workshop, the paper would benefit from a discussion of that topic. If not, please disregard that suggestion.

Response 3: COOPs were minimally discussed during the workshop. However, with the includsion of the NewSTEPs citation, those seeking information about COOPs can seek them out via their website.

Point 4: Figure 1 has a typo in the top section with “N” missing from the word “SPECIMEN”

Response 4: Typo fixed

Point 5: I find it curious that the authors included educating the family in the timeline of one week for the newborn screening process.  I believe it is widely agreed that education should start well before the collection of a specimen and suggest the authors consider a figure revision to reflect the importance of that step earlier in the process.

Response 5: We 100 percent agree that education should start well before the collection of the specimen and the importance of early educaion. Figure 1 is focused on the steps needed to occur within the first week of birth to highlight all the work that must be done within a relatively short amount of time. Within that timeframe, education at the collection of specimen is included. We updated the figure title to highlight that the process dscssed in the figure is what occurs after birth to ensure that readers do not come away thinking that education should only occur when at the collection of specimen.

Point 6: On lines 129-130, the authors use the word “staff” twice in a sentence and it is unclear to which staff they are referring each time.

Response 6: Now on lines 136-137, the staff discussed in that sentence are the short-term follow-up staff as referred to in line 136. Clarification was added to identify that the limited number of staff in that situaiton impacted follow-up visits.

Reviewer 2 Report

No comments

Author Response

Thank you for the review.

Round 2

Reviewer 1 Report

Thank you for your edits. The present form will be important to help NBS programs assess their own responses to the COVID-19 pandemic.